# Long-Term Effects of Repetitive Transcranial Magnetic Stimulation on Tinnitus in a Guinea Pig Model

**DOI:** 10.3390/brainsci12081096

**Published:** 2022-08-18

**Authors:** Farah Amat, Jack W. Zimdahl, Kristin M. Barry, Jennifer Rodger, Wilhelmina H. A. M. Mulders

**Affiliations:** 1The Auditory Laboratory, School of Human Sciences, University of Western Australia, Crawley, WA 6009, Australia; 2School of Biological Sciences, University of Western Australia, Crawley, WA 6009, Australia; 3Perron Institute for Neurological and Translational Research, Crawley, WA 6009, Australia

**Keywords:** tinnitus, guinea pig, repetitive transcranial magnetic stimulation, prefrontal cortex, acoustic trauma, compound action potential thresholds

## Abstract

The auditory phantom sensation of tinnitus is associated with neural hyperactivity. Modulating this hyperactivity using repetitive transcranial magnetic stimulation (rTMS) has shown beneficial effects in human studies. Previously, we investigated rTMS in a tinnitus animal model and showed that rTMS over prefrontal cortex (PFC) attenuated tinnitus soon after treatment, likely via indirect effects on auditory pathways. Here, we explored the duration of these beneficial effects. Acoustic trauma was used to induce hearing loss and tinnitus in guinea pigs. Once tinnitus developed, high-frequency (20 Hz), high-intensity rTMS was applied over PFC for two weeks (weekdays only; 10 min/day). Behavioral signs of tinnitus were monitored for 6 weeks after treatment ended. Tinnitus developed in 77% of animals between 13 and 60 days post-trauma. rTMS treatment significantly reduced the signs of tinnitus at 1 week on a group level, but individual responses varied greatly at week 2 until week 6. Three (33%) of the animals showed the attenuation of tinnitus for the full 6 weeks, 45% for 1–4 weeks and 22% were non-responders. This study provides further support for the efficacy of high-frequency repetitive stimulation over the PFC as a therapeutic tool for tinnitus, but also highlights individual variation observed in human studies.

## 1. Introduction

Tinnitus is a common auditory phantom perception that refers to the conscious perception of sound in the absence of a physical stimulus. Tinnitus is clinically heterogenic, with the intensity and severity of symptoms varying [1,2]. Some patients report not being unduly troubled by their tinnitus and do not seek medical attention, whilst others experience considerable difficulty [1]. Severe side effects of tinnitus, which affect approximately 2 to 3% of the population, include decreased speech discrimination, sleep disturbance and cognitive difficulties–all of which can negatively impact quality of life [3]. Tinnitus is comorbid with anxiety and depression [4,5,6] and the associated psychological distress may increase the risk of suicidal ideation, though evidence is limited for a causal link between the two [6].

While the neural substrate of tinnitus remains unknown, a clear association exists between damage to the peripheral hearing organ, i.e., the cochlea, hearing loss and tinnitus [1,7]. Cochlear damage and hearing loss have been shown to cause central auditory brain changes [8], such as increased neural synchrony and increases in spontaneous and burst firing [9,10,11,12,13], which have been shown to be associated with the presence of tinnitus in multiple animal models [11,13,14] as well as in human studies [15,16]. In addition, tinnitus pathophysiology is not only constrained to auditory sensory regions but also involves non-auditory areas, such as the amygdala, hippocampus and prefrontal cortex (PFC) [8,17,18,19].

In view of the altered neural activity in auditory and non-auditory regions in tinnitus subjects, it is not surprising that many potential therapies are aimed at modulating this activity. One proposed therapeutic intervention is repetitive transcranial magnetic stimulation (rTMS), a non-invasive technique that uses electromagnetic induction to alter the excitability of neurons [20,21], and which is used clinically in the treatment of neurologic and psychiatric disorders [22,23]. rTMS-induced responses depend on physical and biological parameters, including coil characteristics, stimulation pattern, frequency and intensity [24]. Studies in tinnitus patients have shown some success in alleviating tinnitus symptoms with most focusing on the stimulation of the auditory cortex/temporal regions and/or the PFC [25,26,27], although evidence for a beneficial effect remains limited and it is unclear how long effects may persist after the completion of treatment [28,29,30]. A better understanding of the persistence of therapeutic effects of rTMS in the treatment of tinnitus is therefore required and is the focus of this study.

In our laboratory, we investigated the effects of rTMS on tinnitus using a guinea pig model. This animal model allows for more invasive measurements of brain activity and structure and eliminates the placebo effect that is common amongst perceptual disorders such as pain or tinnitus [31,32]. Recently, we have shown therapeutic effects of an excitatory, high-frequency, high-intensity rTMS protocol over the PFC on the behavioral symptoms of tinnitus in our animal model [33]. However, only acute effects were investigated. In the present study, we aimed to investigate the duration of the beneficial effect, measuring the behavioral evidence of tinnitus weekly for 6 weeks after the cessation of rTMS treatment.

## 2. Materials and Methods

### 2.1. Animals and Experimental Design

Experimental procedures were conducted in compliance with the Australian National Health and Medical Research Council code for the use and care of animals and were approved by The University of Western Australia’s (UWA) Ethics Committee (RA/100/1648). Thirteen male (*n* = 8) and female (*n* = 4) Hartley tricolour guinea pigs (Cavia Porcellus), sourced from a breeding colony at UWA, were used in the present study. Guinea pigs weighed between 249 and 402 g (303 ± 11; Mean ± SEM) at the start of testing. Animals were housed in same sex pairs (the 14th animal making up the pairs was part of another experiment) for the duration of the experiment.

The experimental design is summarized in Figure 1. First, the baseline measures for behavioral outcomes were obtained. Then, surgery, under full anesthesia, was performed to assess peripheral auditory thresholds followed by a unilateral acoustic trauma (AT) to induce hearing loss. Following recovery, behavioral tests for tinnitus were performed at weekly intervals. The development of tinnitus was followed by high-frequency and high-intensity rTMS treatment over the prefrontal cortex, administered for ten minutes a day, on weekdays only, over a two-week period. After two weeks of treatment, animals were reassessed for behavioral signs of tinnitus, every week for six weeks. Animals were then euthanized and brains were harvested for immunohistochemical analysis.

Detailed descriptions of behavioral testing for tinnitus [33,34,35,36,37], AT surgery [10,38,39,40] and high-intensity rTMS and immunohistochemistry [33] have been described in previous papers from our laboratory and will only be described briefly in the following sections.

### 2.2. Behavioral Assessments for Tinnitus

During prepulse inhibition (PPI) and gap prepulse inhibition of acoustic startle (GPIAS) tests, animals were mildly restrained in a custom-made polycarbonate holder placed on a force-transducing platform, in a dark soundproof room. Guinea pigs were tested in batches of two (single sex). Before each testing session, guinea pigs were allowed to acclimatize for five minutes. To avoid habitation, testing sessions were never conducted on consecutive days and the order of tests (8 and 14 kHz) were alternated between sessions.

PPI occurs when a relatively weak prepulse stimulus inhibits the startle response to a succeeding stronger stimulus. The PPI test consisted of 50 trials, each lasting 15–30 s (randomized duration), containing a startle stimulus (Radio Shack 401278B; 1 kHz center frequency, 100 Hz bandwidth, 115 dB SPL (Sound Pressure Level), 20 ms duration, 0.1 ms rise/fall time) in silence. Using a separate speaker, a prepulse stimulus (Beyer DT 48; narrowband 8 or 14 kHz center frequency, 3 dB bandwidth = 1 kHz, 66 dB SPL, 50 ms duration) was delivered in half of the trials (randomized presentation), starting 100 ms before the presentation of the startle stimulus.

GPIAS testing is a variant of PPI in which a silent gap, effectively functioning as a prepulse, is inserted in a continuous background noise preceding the startle tone. The GPIAS test was also composed of 50 trials. A startle stimulus (same as PPI) was presented in a continuous background noise (same characteristics as prepulse in PPI), in each trial. Half of the trials contained a 50 ms silent gap, starting 100 ms before the presentation of the startle pulse. Gap and no gap GPIAS trials were randomized as well as the intervals between the presentation of the startle stimulus (15–30 s).

For analysis, the startle response (RMS force) of prepulse trials was compared with no prepulse trials (PPI) or gap trials with no gap trials (GPIAS) within each animal. Mean PPI or GPIAS suppression was expressed in percentage by comparing the startle response between the prepulse and no prepulse trials or between the gap and no gap trials, respectively. An animal was considered to “pass” if there was a significant difference (Mann-Whitney test, *p* < 0.05) between prepulse/no prepulse or gap/no gap trials, and “fail” if the condition is not met (Mann-Whitney test, *p* > 0.05). All animals passed the PPI test once and the GPIAS paradigm twice indicating stable baselines before AT.

Tinnitus development was assessed after AT surgery via weekly GPIAS testing. Animals may fail GPIAS due to tinnitus but also alternatively because of either hearing loss or deficits in the neural circuitry underlying the startle response and PPI. Therefore, animals were required to pass PPI (*p* < 0.05) after AT to ensure that the failure of GPIAS was specifically related to tinnitus. Animals that failed GPIAS (*p* > 0.05) on two repeat occasions (at least 1 day between sessions) and passed PPI were considered to have behavioral signs of tinnitus. For group comparisons, GPIAS suppression was averaged over two sessions before the AT, at the time of tinnitus development and in the weeks after TMS treatment.

### 2.3. Acoustic Trauma Surgery

The anesthesia paradigm consisted of a subcutaneous (sc) injection of Atropine (0.1 mL), followed by Diazepam (intraperitoneal (ip) 0.1 mL/100 g), and 20 min later, an intramuscular (im) dose of Hypnorm (0.315 mg/mL fentanyl citrate and 10 mg/mL fluanisone; 0.1 mL/100 g). The incision site was shaven, and Lignocaine was administered sc (0.1 mL). Following full surgical anesthesia (the absence of foot withdrawal and eyeblink reflex), animals were placed in a soundproof room on a heating blanket and secured into hollow ear bars. If foot withdrawal returned, one-third of the Hypnorm dose was re-administered.

To assess peripheral auditory thresholds, the bulla on the left side was exposed and the round window of the cochlea visualized via a small bullostomy. A silver plastic insulated wire was placed onto the round window to record compound action potential (CAP) thresholds for frequencies ranging from 4 to 24 kHz (2 kHz steps) using a close sound system with a reversed microphone condenser (Bruel and Kjaer, type 4134). Custom software (sample rate 96 kHz, Neurosound; MI Lloyd) generated the sound stimuli with compound action potentials recorded using Scope (Powerlab 4SP, AD Instruments-Australia, Bella Vista, NSW, Australia). Animals were then exposed to a 2 h AT in their left ear (continuous pure tone; 10 kHz at 124 dB), during which the contralateral ear was blocked with plasticine. Following AT, CAP measurements were repeated, the incision sutured and animals were allowed to recover. One week after the acoustic trauma procedure, behavioral testing recommenced for signs of tinnitus.

### 2.4. rTMS Treatment

Only animals that developed tinnitus following AT received rTMS (see Section 2.2 “Behavioral assessments for tinnitus”: failed GPIAS and passed PPI). An animal specific coil (Cool-40 Rat, Magventure, circular; outer diameter 40 mm, Farum, Denmark) coupled with a rTMS power and waveform generator (MagPro R30, Farum, Denmark) was used to deliver rTMS treatment. The coil was placed over the PFC at 14.2 mm anterior to the guinea pigs’ interaural line based on the shape of the induced magnetic field (electromagnetic field and pulse parameters are described in [41]). During stimulation, guinea pigs were awake and gently restrained in the experimenter’s lap to minimize stress. The stimulation protocol was selected to mirror the parameters set by Zimdahl et al. (2021), with machine stimulus output set to 23% to avoid direct facial muscle twitching. High-frequency, high-intensity rTMS (2000 pulses at 20 Hz) was delivered for two weeks, for 10 min a day, during weekdays only (Monday to Friday). To prevent overheating the inter-interval train was set to 13 s long. Following the treatment, animals were tested for tinnitus every week for 6 weeks (see Section 2.2 Behavioral assessments for tinnitus). The first tinnitus test took place on the Monday following the cessation of rTMS treatment (Friday).

### 2.5. End of Experiment

After the last tinnitus tests at 6 weeks post-treatment cessation, animals were anesthetized and CAP thresholds measured on both sides as described in Section 2.3 “Acoustic trauma surgery”. Then, animals were euthanized with a barbiturate overdose (ip injection of Lethobarb 0.4 mL (Pentobarbital; 325 mg/mL)) followed by transcardiac perfusion with saline and 4% paraformaldehyde in 0.1 M phosphate buffer (PB). Brains were removed, stored overnight in 4% paraformaldehyde in PB and then immersed in 30% sucrose in PB for at least 48 h at 4 °C.

### 2.6. Immunohistochemistry and Analysis

Brain sections (60 µm) were cut on a cryostat. One in every seven sections was then incubated overnight (4 °C) in mouse anti-calbindin (1:500, Sigma-Aldrich, St Louis, MO, USA) in 0.1 M phosphate buffer, 0.1% albumin bovine, 0.3% Triton and 5% donkey serum. This was followed by incubation for 90 min (at room temperature), in donkey anti-mouse (1:500, Merck KGaA, Darmstadt, Germany) in 0.1 M phosphate buffer, 0.1% albumin bovine, 0.3% Triton and 5% donkey serum, followed by incubation in avidin-biotin complex (1:800 A and B in 0.1 M phosphate buffer; 90 min at room temperature). 3,3′-Diaminobenzidine was used as a chromogen to visualize the staining. Sections were mounted on gelatinized slides, dehydrated and covered with Entellan (Merck KGaA, Darmstadt, Germany).

Immunopositive neurons were then counted in both hemispheres using a counting frame (0.24 mm^2^), on NIS Elements software. Images and photomicrographs were captured using a Nikon Eclipse 80i (×20 objective lens) and an integrated Digital Sight Camera (NIS-elements Basic Elements software, Nikon Australia, Rhodes, NSW, Australia). Since we showed in our previous study [33] that rTMS only altered the densities of calbindin in the dorsal areas (close to skull) of the prefrontal cortex, neuronal densities of calbindin were analyzed in two dorsal regions of interest of the prefrontal cortex. The soma and two dendrites of an immunopositive cell had to be visible in order to be counted. For details on the regions of interest and calculations of densities see [33].

### 2.7. Analyses

A one-way analysis of variance (ANOVA) with repeated measures was used to analyze both behavioral (GPIAS) and CAP threshold data. This was followed up by appropriate post hoc tests when required (e.g., Sidak’s multiple comparisons). A Pearson’s correlation of time after treatment and gap suppression (as determined from the GPIAS test) was also performed. Skewness and kurtosis statistics paired with a Shapiro–Wilk test were used to confirm normality. Neuronal densities were compared with data from a previous study [33] from sham treatment and early time point (3 days) after the cessation of rTMS treatment.

## 3. Results

### 3.1. Hearing Loss

CAP thresholds in the left (experimental ear) of all animals (*n* = 13) before (baseline) and immediately after AT (post-AT) as well as at the end of experiment (EOE) are shown in Figure 2A. The graph shows the immediate large effects of AT and the partial recovery of thresholds at EOE. Due to a technical issue, we were unable to record CAP thresholds in one animal at EOE; thus, a mixed model ANOVA was used for analysis. Mixed effect analysis showed significant effects for time F (1.917, 242.5) = 605.3. Post hoc comparisons of post-AT with the baseline values showed significant (*p* < 0.01) effects immediately after AT at all frequencies with the exceptions of 4 kHz. At EOE, CAP thresholds were significantly elevated compared to the baseline at 8, 10, 12 and 16 kHz (*p* < 0.05) and 18 to 24 kHz (*p* < 0.01).

CAP thresholds at EOE in the control (right) ear were not significantly different from the left ear baseline (Figure 2B) (multiple *t*-tests with correction), showing that the AT on the left ear did not have any effects on the right-hand side. CAP threshold loss was also compared between males (*n* = 8) and females (*n* = 4) and although the CAP threshold loss seemed to be elevated in females (Figure 2C) these differences failed to reach or approach statistical significance (multiple *t*-tests with correction).

### 3.2. Tinnitus Development

Out of the 13 animals that underwent AT surgery, 10 (77%) developed behavioral signs of tinnitus between 13 and 60 days (34 ± 4; Mean ± SEM) after the AT. Five of these ten animals developed a failed GPIAS at 8 kHz, 3 at 14 kHz and two at both frequencies. This variety in frequency at which the animals develop tinnitus is in agreement with our previous studies showing that GPIAS failure can occur at either frequency or both frequencies after an AT, as used in this study [33,35,37,42,43]. The background noise/prepulse center frequencies used for GPIAS and PPI (8 and 14 kHz) were selected as these frequencies represent an audiogram region without threshold loss (8 kHz) and a region showing threshold loss (14 kHz) following our AT paradigm. Tinnitus developed in all females (*n* = 4) and in 6 of the 9 males.

There was no significant difference in CAP threshold loss at EOE between tinnitus (*n* = 9) and non-tinnitus (*n* = 3) animals (multiple t-tests with correction) (Figure 3A). A Pearson correlation analysis showed no relationship (*p* = 0.636) between CAP threshold loss at EOE (summed over all frequencies) and days to develop tinnitus (Figure 3B). There was also no significant difference in immediate CAP threshold loss between tinnitus (*n* = 9) and non-tinnitus (*n* = 3) animals (multiple t-tests with correction) and a Pearson correlation analysis showed no relationship (*p* = 0.447) between immediate CAP threshold loss (summed over all frequencies) and days to develop tinnitus (data not shown).

### 3.3. Effect of Treatment

Nine of the ten animals underwent the rTMS treatment. In one animal (male), the treatment was ceased as this animal did not remain still on the lap of the experimenter using mild restraint making the placement of the coil unreliable. Although a stronger restraint was considered, this was deemed to potentially induce additional stress and hence dismissed as a solution. The other animals tolerated rTMS well, with no adverse side effects observed.

For the analysis of the GPIAS data before and after rTMS treatment, the percentage of GPIAS suppression at the frequency of tinnitus development was compared using a one-way repeated measures ANOVA followed by Sidak post hoc tests. For the two animals that developed GPIAS failures at both 8 and 14 kHz, the average GPIAS for both frequencies was calculated. Figure 3C shows the average percentage GPIAS suppression at the tinnitus and non-tinnitus frequency at baseline, at the time point of tinnitus development post-AT and then at 6 weeks after the completion of rTMS treatment. A one-way repeated measures ANOVA showed a significant effect of time (F (3.408, 27.26) = 3.957, *p* = 0.0151). Post hoc Sidak tests between the pre-AT time point and the other time points revealed that GPIAS suppression was significantly reduced at the post-AT tinnitus time point (*p* = 0.0057) but GPIAS suppression at week 1 to week 6 following the rTMS treatment showed no significant differences with the baseline time point. Further post hoc Sidak tests comparing the post-AT tinnitus time point with the weekly time points after treatment showed a significant effect with week 1 (*p* = 0.0394) but no significance with the other time points. These data show that rTMS treatment resulted in the clear attenuation of tinnitus in the group of animals at week 1 post-treatment but the results from week 2 to 6 seem more ambiguous and difficult to interpret. As can be observed in Figure 3C, results at these time points are neither significantly different from the no tinnitus (pre-AT) or tinnitus (post-AT) time point, most likely due to the large inter-animal variability.

A further analysis of individual data was performed and the data are shown in Figure 4. Qualitatively, animals could be tentatively separated into three groups, i.e., responders (*n* = 3), partial responders (*n* = 4) and non-responders (*n* = 2). The animals in the responders group exhibited sustained remission over the 6 weeks. Partial responders were sensitive to treatment initially (showing sustained beneficial effects between 1–4 weeks); however, their tinnitus returned over time. Non-responders did not show direct effects of treatment and demonstrated variability in tinnitus symptoms that did not seem to be correlated to treatment.

No apparent relationship of responsiveness to treatment with sex was apparent between the groups. Average time post-AT in days of tinnitus development seemed higher in the responders group (48 ± 6.25) compared to the partial responders (26.3 ± 5.62) and non-responders (24.5 ± 3.5) but a one-way ANOVA failed to reach significance (*p* = 0.0588) and this analysis should be interpreted with caution, in view of the low numbers of animals in each subgroup.

### 3.4. Neuronal Density of Calbindin Postivive Neurons in PFC

The neuronal densities of calbindin positive neurons in PFC in RO1 (most dorsal region) and RO2 were compared with data previously obtained in another study from our laboratory [33] using unpaired *t*-tests. Comparisons were made with a group of sham rTMS-treated animals (*n* = 6) and a group treated with identical rTMS settings (high frequency, high intensity over PFC) but analyzed 3 days after the cessation of treatment (*n* = 6). As we have shown previously, soon after rTMS treatment cessation (3 days), the densities of calbindin-positive neurons increased (*p* = 0.0383) compared to the sham treatment in ROI1 only, but this increase was no longer visible 6 weeks after the cessation of rTMS treatment (present study) (Figure 5), suggesting that the effects on calbindin expression in the PFC were not long lasting.

Next, we investigated a possible correlation between the calbindin density in the animals of the current study with firstly the duration of therapeutic effects and secondly with the time it took for animals to develop tinnitus after AT. No correlation was found in either ROI between neuronal density and the duration of therapeutic effect on tinnitus (data shown for ROI1, Figure 5C), which suggests that the variability in the duration of treatment effect may not be associated with the duration of effects on calbindin-positive neuron density in PFC. However, a correlation was found between the time it took for animals to develop tinnitus after AT and calbindin density six weeks after the end of treatment (Figure 5B; Pearson correlation analysis *p* = 0.0457). This correlation suggests that in animals that developed tinnitus faster, the effects of rTMS on the density of calbindin-positive neurons were longer lasting.

## 4. Discussion

The present study confirms the results from Zimdahl et al. (2021) [33], which indicated that a 2-week course of daily (10 min, weekdays only) 20 Hz, high-intensity rTMS treatments over PFC attenuates the signs of tinnitus in the first week following the completion of treatment in a guinea pig model. In addition, it shows that the effects can last up to 6 weeks in some animals. Interestingly, the data also show considerable variability in the duration of treatment effects in agreement with what is observed in the human population, particularly in patients receiving rTMS treatment for depression [30]. This highlights the need to be able to stratify tinnitus populations not only to optimize therapeutic outcomes, but also to identify the optimal regime for maintenance treatments to sustain therapeutic effects as long as possible.

As hearing loss and noise overexposure are significant risk factors for the development of tinnitus, it is not surprising that many animal models use acoustic trauma to induce tinnitus development [44]. These models reflect human clinical findings both with regard to the effects on cochlear thresholds and tinnitus development as shown in the current data. Acoustic trauma resulted in an immediate threshold shift at multiple frequencies, with the largest effects at frequencies above the acoustic trauma frequency. This pattern of decreased hearing sensitivity is consistent with the well described phenomenon of the half-octave shift [45] and other animal models of induced hearing loss [46,47,48] and clinical findings [49,50,51,52]. As expected [52], the large threshold shift was only temporary with CAP thresholds showing considerable recovery at the end of the experiment, revealing a less severe permanent threshold shift. This permanent change in thresholds is likely to result from cochlear hair cell damage and/or loss [52,53,54]. Following the acoustic trauma, a total of ten out of thirteen animals (77%) developed a GPIAS deficit, whilst still passing the PPI test, consistent with behavioral signs of tinnitus [54,55]. Other studies using acoustic trauma in animal models report percentages of animals developing tinnitus between 30 and 70% [13,33,56,57,58]. This variance may be due to experimental differences, such as the choice of species, intensity and/or duration of acoustic trauma and whether the timeline of the experiment allowed all tinnitus development to be captured [38,44]. In human populations, the reported prevalence of tinnitus in hearing loss populations shows variation between 30 and 90%, with prevalence suggested to be dependent on the etiology of the hearing loss [59,60,61,62,63]. In the present study, threshold loss was not different between tinnitus and non-tinnitus animals which is in line with human studies showing that the magnitude of hearing loss is not a predictor of tinnitus development [7,64].

Because the aim of the present study was to investigate the persistence of beneficial effects of rTMS, we did not include a sham treatment but employed a longitudinal design. This means the possibility cannot be ruled out that spontaneous recovery may have occurred in some animals. However, our previous study [33] showed a clear beneficial effect of rTMS against sham and all of the responders in the present study showed an attenuation of tinnitus at the completion of treatment and not at random time points in the recovery period of 6 weeks. Taken together, this suggests a genuine long-lasting effect of rTMS in the majority of animals.

The amelioration of tinnitus observed in week 1 following the completion of rTMS treatment is in agreement with our previous study in guinea pigs showing the effectiveness of high-frequency stimulation over PFC [33] as well as a recent study in humans by Ciminelli and colleagues [26]. The latter study showed a significant improvement in the Tinnitus Handicap Inventory that lasted for up to 16 weeks (longest time point investigated in the study). The authors also described a reduction in tinnitus loudness though this just failed to reach significance. Other studies in humans targeting PFC with high-frequency TMS used combined paradigms, stimulating the temporal cortex as well as PFC, and also reported beneficial effects [25,27,65,66,67]. Interestingly, a study by Noh et al. [68] that applied low-frequency stimulation to PFC described a limited effect, suggesting that the activation rather than the inhibition of PFC is required for therapeutic effects on tinnitus.

Although our data show that rTMS had long-term benefits in a subset of animals and large variability was seen at the individual level. Three of the animals showed the attenuation of tinnitus for 6 weeks, four animals had a transient improvement between 1 and 4 weeks, but two animals did not respond at all. This pattern reflects the variability in individual responses in human patients after high-frequency stimulation of the PFC despite a beneficial effect for the whole cohort [26]. Other studies, which used combined stimulation paradigms of prefrontal and temporal regions in tinnitus patients also report both responders and non-responders to treatment with no relationship to sex as shown in the present study [69,70]. Our data did suggest that animals that took longer to develop tinnitus were more likely to respond to treatment, but this has, to our knowledge, not been explored in human studies. A study by Yang and colleagues [70] suggests that subjects with recent onset are more likely to respond to rTMS treatment, but this was not explored in our study as treatment started within a week after tinnitus was observed. Hence, it remains to be investigated why some animals responded to treatment and others did not, as so often observed in human studies. This could for example involve the measurement of stress levels or neural hyperactivity and its underlying mechanisms. This would increase our understanding of the mechanisms of tinnitus and would potentially allow the stratification of the population, leading to a more tailored treatment to the individual.

The mechanisms by which rTMS over PFC attenuates tinnitus remains to be resolved. As discussed in our previous study [33], an excitatory effect on PFC from the high-frequency stimulation [71,72] may result in direct or indirect activation of cells in the thalamic reticular nucleus [73,74] which in turn would provide an inhibitory effect on medial geniculate nucleus activity [75] and consequently lower activity in auditory cortex. Alternatively, the activation of PFC by high-frequency rTMS may lead to changes in the auditory cortex via direct pathways independently of the medial geniculate nucleus [76,77,78]. Our immunohistochemical results on the density of calbindin-positive neurons in PFC showed that the increased density seen at the early time point following the cessation of rTMS [33] was not sustained after 6 weeks. The lack of long-term changes in density is most simply interpreted as there being no long-term changes in calbindin circuitry, although we cannot rule out that there were increases and decreases in expression within different cells that did not affect overall counts, and thus were not detected by our relatively simple approach of quantifying density at one time point. Calbindin density at 6 weeks after the end of treatment was not correlated with the effectiveness of treatment on tinnitus, though there was a significant correlation with the time it took to develop tinnitus. A possible interpretation is that PFC activity/involvement may be different in animals that develop tinnitus fast compared to animals that develop tinnitus slower. The fact that higher calbindin density at the 6-week time point was correlated with a shorter time taken to develop tinnitus is in apparent conflict with our data showing that animals that took longer to develop tinnitus were better responders to treatment. However, there is no direct evidence that the change in calbindin density in PFC by rTMS is causative of the suppression of tinnitus. In addition, an alternative interpretation is that animals that develop tinnitus faster respond to rTMS with a more sustained increase in calbindin density which may, in effect, not be beneficial to tinnitus outcomes and a transient increase in calbindin density may be preferred. Studies investigating intermediate time points between the 3 days and 6 weeks explored by us, may shed light on this possibility.

## 5. Conclusion

In conclusion, our data support the therapeutic use of high-frequency rTMS stimulation over PFC for individuals with tinnitus. In addition, our data suggest that effects may be long lasting in some individuals, but others may benefit from ongoing maintenance treatments to sustain therapeutic effects [30]. It is important to note that an animal study is not affected by possible placebo effects, since an animal would not be experiencing the perceptions and expectations that a human tinnitus patient would have when receiving treatment.

## Figures and Tables

**Figure 1 brainsci-12-01096-f001:**
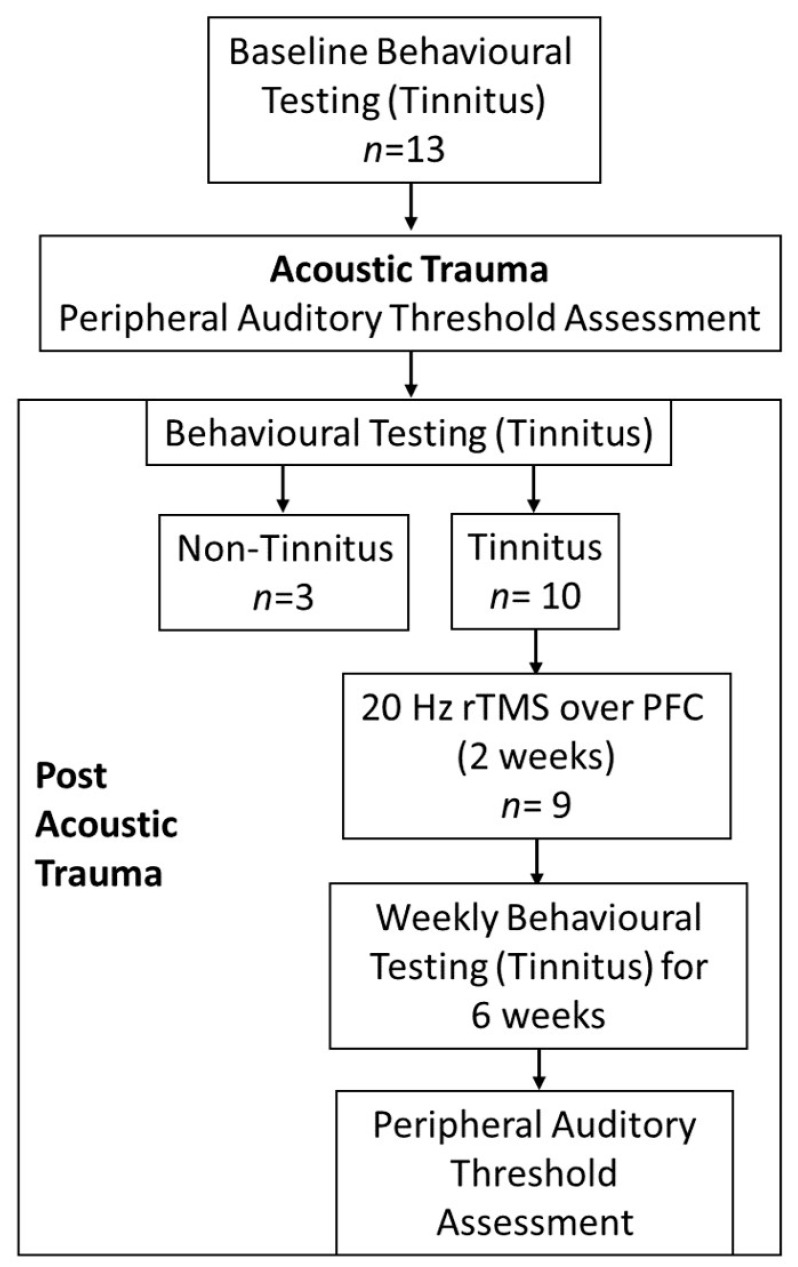
Experimental flow diagram. First, baseline measures for behavioral outcomes (tinnitus testing) were obtained. Then, surgery, under full anesthesia, was performed to assess peripheral auditory thresholds followed by a unilateral acoustic trauma (AT) to induce hearing loss. Following recovery, behavioral tests for tinnitus were performed at weekly intervals. Development of tinnitus was followed by high-frequency and high-intensity rTMS treatment over the prefrontal cortex, administered for ten min a day, on weekdays only, over a two-week period. After two weeks of treatment, animals were reassessed for behavioral signs of tinnitus, every week for six weeks. Animals were then euthanized and brains harvested for immunohistochemical analysis.

**Figure 2 brainsci-12-01096-f002:**
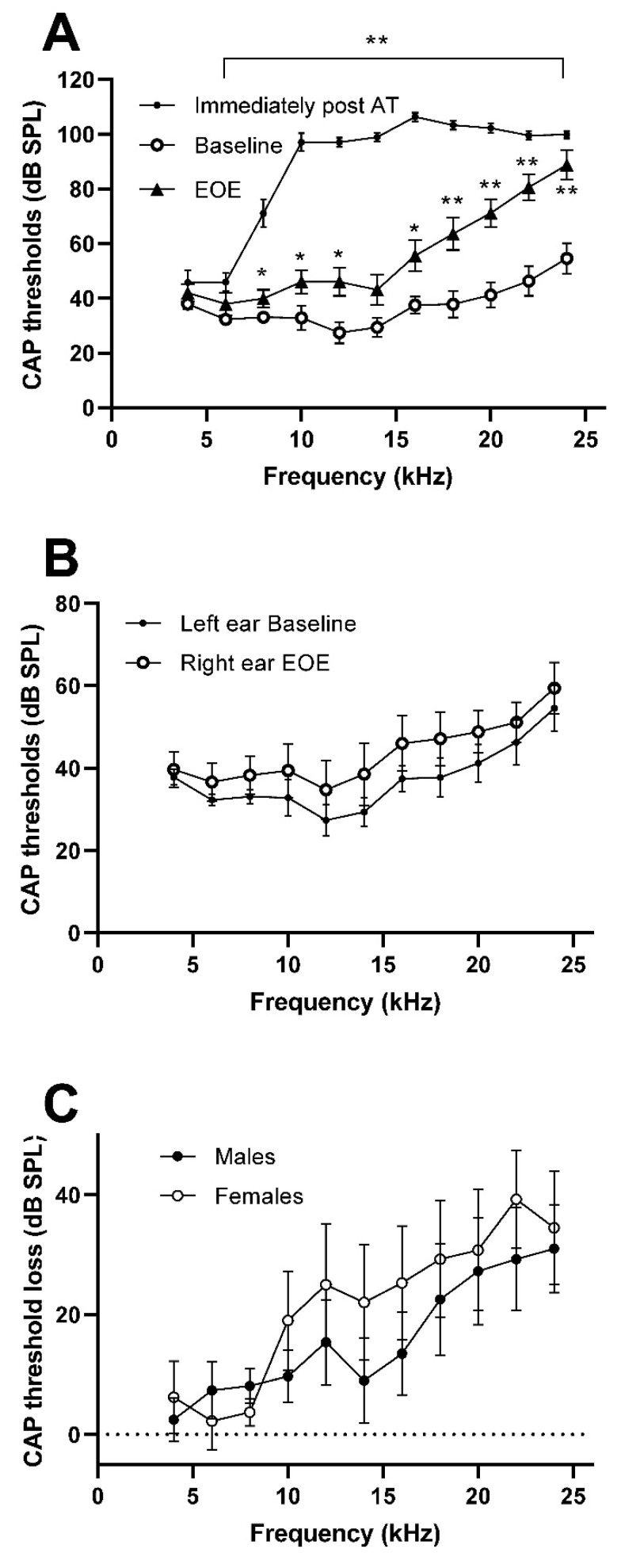
Graphs showing CAP thresholds (dB SPL) against frequency to demonstrate the effect of AT. (**A**): Before AT (baseline; *n* = 13), immediately after the AT (*n* = 13) and at the end of experiment (EOE; *n* = 12). (**B**): Left ear (AT side) before AT and right ear at EOE, showing normal thresholds (*n* = 13). (**C**): EOE thresholds of males (*n* = 9) and females (*n* = 4). All mean ± SEM. * *p* < 0.05; ** *p* < 0.01.

**Figure 3 brainsci-12-01096-f003:**
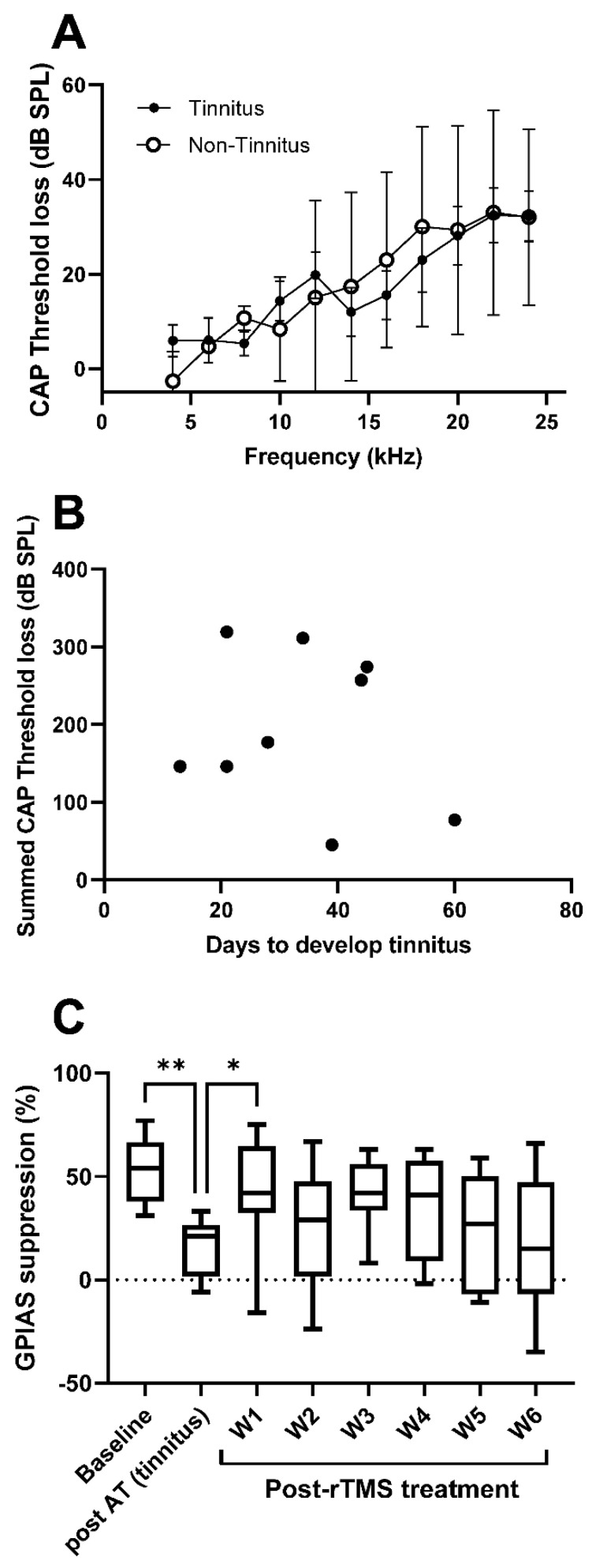
(**A**): Graph showing no difference in CAP thresholds (dB SPL) in tinnitus (*n* = 9) and non-tinnitus (*n* = 3) animals at EOE. (**B**): Scatterplot showing lack of correlation between CAP threshold loss (summed over all frequencies) and days it took to develop tinnitus in each animal. (**C**): Box and whisker plot of GPIAS suppression (%) before AT (baseline), at the time of tinnitus development (post-AT) and at week 1 to 6 post-treatment. * *p* < 0.05; ** *p* < 0.01.

**Figure 4 brainsci-12-01096-f004:**
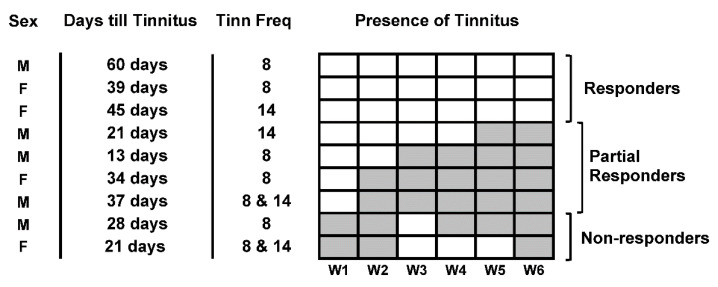
Figure showing absence (white box) or presence (grey box) of tinnitus for each animal in the 6 weeks post-treatment, illustrating that 3 animals showed sustained absence of tinnitus (responders), 4 animals that showed a gradual return of tinnitus (partial responders) and 2 animals in which treatment had no apparent effect (non-responders). Graph also indicates sex, days to develop tinnitus and tinnitus frequency (kHz) for each animal.

**Figure 5 brainsci-12-01096-f005:**
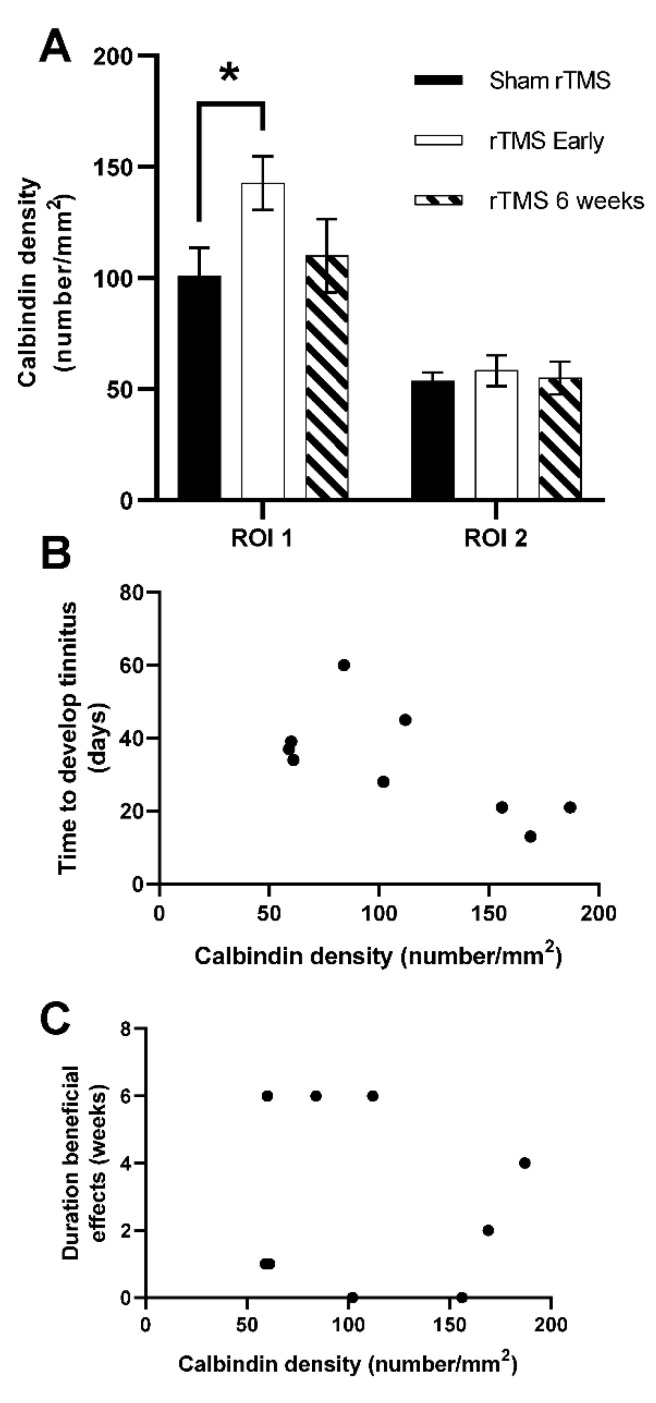
Data on density of calbindin-positive neurons. (**A**): Bar graph showing densities of calbindin-positive neurons in two regions of interest (RO1 and RO2) in PFC comparing the result from the present study, 6 weeks after the cessation of treatment (rTMS 6 weeks) with data collected previously from sham rTMS treatment and soon after rTMS treatment (3 days after the cessation of treatment) [33]. A significant upregulation soon after rTMS treatment (*p* = 0.0383; indicated with *) was observed only in the most dorsal region (RO1) but densities returned to sham levels 6 weeks after cessation of treatment (current data). (**B**): Scatterplot showing significant correlation from data in current study between calbindin density and time it took to develop tinnitus (Pearson correlation analysis, *p* = 0.0457). (**C**). Scatterplot from data in the current study showing lack of correlation between calbindin density and duration of beneficial effects on tinnitus (Pearson correlation analysis; *p* = 0.7717).

## Data Availability

Not applicable.

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
