# Peer review of "Long-Term Effects of Repetitive Transcranial Magnetic Stimulation on Tinnitus in a Guinea Pig Model"

_brainsci, 2022, doi:10.3390/brainsci12081096_

Round 1

Reviewer 1 Report

This article investigated high-frequency, high-intensity rTMS in a tinnitus animal model, aiming to provide evidence of the effectiveness of rTMS in treating acoustic trauma related tinnitus. Such animal model can successfully lead to tinnitus and quantify the severity of tinnitus, therefore provide a good approach to investigate different brain stimulation protocols in treating tinnitus.

1. Even though animal study is not affected by placebo effects, a control group is still essential. Before concluding the benefit of rTMS treatment, it is important to elucidate whether tinnitus amelioration is spontaneous or resulted from rTMS treatment.

2. In Figure 3A and 3B, the authors demonstrate CAP threshold in tinnitus/ non-tinnitus group and the correlation between CAP threshold and days to develop tinnitus. It is wondering why CAP threshold at “EOE” was analyzed but not CAP threshold “immediate after AT”?  

3. Does GPIAS suppression failure at 8 or 14 kHz has different pattern of CAP threshold loss or different response to rTMS treatment?

Author Response

  1. Even though animal study is not affected by placebo effects, a control group is still essential. Before concluding the benefit of rTMS treatment, it is important to elucidate whether tinnitus amelioration is spontaneous or resulted from rTMS treatment.

This paper is a follow up from a previous study in our laboratory in which we compared high-frequency, high-intensity rTMS as used in this study to sham stimulation and showed that this experimental protocol significantly reduced tinnitus compared to sham at an early time-point after cessation of tinnitus (3 days later)  (Zimdahl et al. 2021, Excitatory Repetitive Transcranial Magnetic Stimulation Over Prefrontal Cortex in a Guinea Pig Model Ameliorates Tinnitus. Front Neurosci. 2021 Jul 22;15:693935.). The purpose of the current paper was to investigate whether the beneficial effects would change over time as the literature is unclear regarding the duration of beneficial effects of rTMS. The longitudinal/ repeated measures design of this study, with no control required, allowed us to address the changes over time which is the focus of this paper. We have made changes to introduction and discussion to re-iterate the focus of our paper and the rational.

  1. In Figure 3A and 3B, the authors demonstrate CAP threshold in tinnitus/ non-tinnitus group and the correlation between CAP threshold and days to develop tinnitus. It is wondering why CAP threshold at “EOE” was analyzed but not CAP threshold “immediate after AT”?  

We used permanent threshold loss as we thought this was more relevant as immediate threshold loss is only present for some days and, permanent threshold loss shows more variability than the immediate threshold loss. However, following the reviewer’s comment we have now also performed an analysis using CAP thresholds “immediately after AT”. There was no significant difference between tinnitus and non-tinnitus animals with regards to immediate threshold loss at any frequency and no correlation between immediate CAP threshold loss and days to develop tinnitus (Pearson correlation analysis) (information added to Results ).

  1. Does GPIAS suppression failure at 8 or 14 kHz has different pattern of CAP threshold loss or different response to rTMS treatment?

We have added the tinnitus frequency to figure 4 to inform the reader as to the potential effect on response to treatment, however, as subdividing the animals with tinnitus that went through to treatment results in subgroups containing only 2 or 3 animals we deem a statistical analysis not meaningful.

We did not observe a relationship between the shape of hearing loss and tinnitus frequency. This is in line with our previous work, where we found no clear relationship between the pattern or magnitude of hearing loss and tinnitus frequency (Mulders et al. 2019 Low-intensity repetitive transcranial magnetic stimulation over prefrontal cortex in an animal model alters activity in the auditory thalamus but does not affect behavioural measures of tinnitus, Exp. Brain Res., 237 and Mulders et al. 2016 The effects of repetitive transcranial magnetic stimulation in an animal model of tinnitus Sci. Rep., 6.). 

Reviewer 2 Report

Overall, the manuscript is well structured and written. 

Although the goal was clear, the study and results/interpretation could use improvement and further experimentation. 

Some things to highlight:

I believe Fig.3A is the CAP threshold loss? Otherwise, thresholds do not match EOE data in Fig. 2A. Please confirm. 

The authors use the criteria of pass/fail by statistically comparing prepulse to no prepulse conditions in the GPIAS and PPI. However, in their results (Fig. 3c), they perform analyses on the GPIAS % suppression over the extended time period. Suggestion is to be consistent in what you are defining as behavioral signs of tinnitus and continue to use the pass/fail as shown in Figure 4.  

Of interest to the reader and what would round out this study is to understand the morphological or physiological changes in responders vs. non-responders. The authors point out that previous studies have looked at the longitudinal effect of rTMS in humans. The primary advantage of looking into this in animals is to understand underlying neurological changes and factors that would contribute to a long term response. 

Given that previous studies have shown the effects of rTMS in guinea pigs (including a previous paper from the authors) and in humans, I unfortunately don't see this work as a standalone manuscript in this state. Further experimentation is needed to shed light in the variability observed in this study. 

Author Response

Comment: I believe Fig.3A is the CAP threshold loss? Otherwise, thresholds do not match EOE data in Fig. 2A. Please confirm. 

Reply: Our apologies this was an error. Thank you for picking this up. We have now included a correct version of the figure (Y axis had to read CAP threshold loss not CAP threshold).

Comment: The authors use the criteria of pass/fail by statistically comparing prepulse to no prepulse conditions in the GPIAS and PPI. However, in their results (Fig. 3c), they perform analyses on the GPIAS % suppression over the extended time period. Suggestion is to be consistent in what you are defining as behavioral signs of tinnitus and continue to use the pass/fail as shown in Figure 4.  

Reply: The percentage of GPIAS suppression is strongly correlated to pass/fail. In addition, within the literature some authors use percentage only and some pass/fail criteria. That is the reason why we show both: we show in figure 4 the weekly pass/fail result and in figure 3 we show the GPIAS suppression in percentage.

Comment: Of interest to the reader and what would round out this study is to understand the morphological or physiological changes in responders vs. non-responders. The authors point out that previous studies have looked at the longitudinal effect of rTMS in humans. The primary advantage of looking into this in animals is to understand underlying neurological changes and factors that would contribute to a long term response. 

Reply: The reviewer is of course correct that animal models allow an investigation of underlying mechanisms but a detailed analysis of physiological and morphological parameters was beyond the scope of the present paper. It was important for us to first validate and define the persistence of treatment effects so that we could design future studies with the most appropriate timepoints to investigate potential mechanisms underlying variability and longevity of effects.

However, we did harvest the brains of the animals and have now included an immunohistochemical analysis of calbindin density in PFC. In our previous paper (Zimdahl et al. 2021) we showed that the density of calbindin is increased after rTMS and so we used this marker to evaluate whether the effects on PFC were long-lasting and whether this density (potentially a biomarker for altered state in PFC) was correlated with the variable effects seen on tinnitus. We show that the density of calbindin positive neurons in PFC was not different from sham treatment (compared with results from our previous study) and no correlation existed between treatment effects and levels of calbindin density. This suggests that the variability in duration of treatment effect may not be associated with duration of effects in PFC. However, interestingly there was a correlation between the time it took for animals to develop tinnitus after AT and calbindin density (fig. 5B; Pearson correlation analysis p=0.0457). This suggests that in animals that developed tinnitus faster, effects of rTMS on density of calbindin-positive neurons were longer lasting.

Comment: Given that previous studies have shown the effects of rTMS in guinea pigs (including a previous paper from the authors) and in humans, I unfortunately don't see this work as a standalone manuscript in this state. Further experimentation is needed to shed light in the variability observed in this study. 

Reply: We respectfully disagree with the reviewer. There is very little understanding in humans of when effects of rTMS cease, even though it is clear from patient data that effects wear off over time. Our study is therefore highly important in the context of understanding the duration of therapeutic effects, and will inform recommendation and timing of maintenance treatments. Our previous work used MRI in rats to show persistence of effects up top 2 weeks post end of treatment (Seewoo Brain Stimulation 2019) and this is the first study to follow up on that in a disease (tinnitus) model. Nonetheless, as described above, we have now added an immunohistochemical analysis in PFC (resulting also in additional figure).

Round 2

Reviewer 1 Report

Thanks for providing further information regarding the different clinical presentation among GPIAS suppression failure at 8 or 14 kHz and the relation between tinnitus and CAP threshold immediate after AT.

However, I am still not quite convinced that a longitudinal/ repeated measures design requires no control group. In the previous study, tinnitus was assessed three days after the cessation of treatment and showed that rTMS could significantly reduce tinnitus as compared to sham condition. This results established the early effect of rTMS. However, this results can not guarantee rTMS outperform sham at 6 weeks later. Spontaneous recovery or recovery because of any other systemic reason is possible. Therefore, tinnitus reduction 6 weeks after rTMS treatment could not claim to be attributed to rTMS completely when there’s no control/ sham condition. 

Author Response

We have added some further discussion related to this comment in the Discussion: “Because the aim of the present study was to investigate the persistence of beneficial effects of rTMS, we did not include a sham treatment but employed a longitudinal design. This means the possibility cannot be ruled out that spontaneous recovery may have oc-curred in some animals. However, our previous study [33] showed a clear beneficial effect of rTMS against sham and all of the responders in the present study showed an attenua-tion of tinnitus at completion of treatment and not at random timepoints in the recovery period of 6 weeks. Taken together this suggests we do see a genuine long-lasting effect of rTMS in the majority of animals.”

Reviewer 2 Report

Thank you for the addition of the PFC calbindin density analysis. I appreciate the authors' efforts to provide further insight of what can be driving the individual differences. The correlation seems to be a bit perplexing. How do you tie that in with the behavioral trend that suggest animals that took longer to develop tinnitus may show sustained response from rTMS treatment? 

I would like to ask that the authors perhaps provide some additional thoughts on this finding rather than just stating that the circuitry maybe different. 

Author Response

We agree that there is an apparent conflict between the data in figures 4 and 5. We have now extended the discussion on this issue as follows “The fact that higher calbindin density at the 6 weeks timepoint was correlated with a shorter time taken to develop tinnitus is in apparent conflict with our data showing that animals that took longer to develop tinnitus were better responders to treatment. However, there is no direct evidence that the change in calbindin density in PFC by rTMS is causa-tive of the suppression of tinnitus. In addition, an Further studies are required to explore these mechanisms.alternative interpretation is that animals that develop tinnitus faster respond to rTMS with a more sustained increased in calbindin density which may, in ef-fect, not be beneficial to tinnitus outcomes and a transient increase of calbindin density may be preferred. Studies investigating intermediate time-points between the 3 days and 6 weeks explored by us, may shed light on this possibility.”